# Near-Infrared Time-Resolved Spectroscopy Shows Anterior Prefrontal Blood Volume Reduction in Schizophrenia but Not in Major Depressive Disorder

**DOI:** 10.3390/s22041594

**Published:** 2022-02-18

**Authors:** Toshikazu Shinba, Nobutoshi Kariya, Saori Matsuda, Makoto Arai, Masanari Itokawa, Yoko Hoshi

**Affiliations:** 1Department of Psychiatry, Shizuoka Saiseikai General Hospital, Shizuoka 422-8527, Japan; 2Autonomic Nervous System Consulting, Shizuoka 422-8527, Japan; 3Maynds Tower Mental Clinic, Tokyo 151-0053, Japan; info@mmc-mcc.com (N.K.); matsuda@mmc-mcc.com (S.M.); 4Schizophrenia Research Project, Tokyo Metropolitan Institute of Medical Science, Tokyo 156-8506, Japan; arai-mk@igakuken.or.jp (M.A.); itokawa-ms@igakuken.or.jp (M.I.); 5Department of Biomedical Optics, Institute for Medical Photonics Research, Hamamatsu University School of Medicine, Hamamatsu 431-3192, Japan; yhoshi@hama-med.ac.jp

**Keywords:** near-infrared time-resolved spectroscopy, prefrontal cortex, hemoglobin concentration, brain blood volume, schizophrenia, depression

## Abstract

Previous studies using various brain imaging methods have reported prefrontal blood flow disturbances in psychiatric disorders, including schizophrenia and major depressive disorder. In both disorders, alterations of the resting blood flow, in addition to that of the activation in response to task load, have been shown, but the results are not consistent. The present study aimed to examine the anterior prefrontal hemoglobin concentration at the resting state in schizophrenia and depression using near-infrared time-resolved spectroscopy (NIR-TRS), which estimates the optical absorption coefficients and calculates the absolute concentrations of oxygenated (oxy-Hb), deoxygenated (deoxy-Hb), and total (total-Hb; sum of oxy-Hb and deoxy-Hb) hemoglobin. Their ratios to systemic blood hemoglobin concentration (blood-Hb) were also assessed. In agreement with our previous data, total-Hb and total-Hb/blood-Hb in schizophrenia were significantly lower. The present study further revealed that both oxy-Hb/blood-Hb and deoxy-Hb/blood-Hb in schizophrenia were reduced. In depression, total-Hb, total-Hb/blood-Hb, oxy-Hb, and oxy-Hb/blood-Hb were higher than in schizophrenia and were not different from the control. The oxygen saturation (oxy-Hb/total-Hb), in addition to the optical pathlengths, did not show group differences. Lowered oxy-Hb/blood-Hb and deoxy-Hb/blood-Hb together with unchanged oxygen saturation may indicate that the prefrontal blood volume is reduced in schizophrenia. The present findings suggest that NIR-TRS is useful in analyzing the hemodynamic aspects of prefrontal dysfunction in schizophrenia and differentiating schizophrenia from depression.

## 1. Introduction

It has been revealed that the prefrontal cortex is involved in various brain functions, including the planning, decision, and execution of behavior, and serves to associate information in the human brain [1]. Damage to the prefrontal cortex leads to a wide range of cognitive, emotional, and behavioral changes, such as inattention, apathy, loss of inhibitory control, and deficit in the working memory [1]. It is also indicated that the prefrontal dysfunctions are related to psychiatric disorders, including schizophrenia and depression [2,3]. The former manifests delusion, hallucination, and thought disorder, and the latter depressed feeling and anhedonia [4]. Both disorders are symptomatically different but involve disturbed information processing in the prefrontal cortex [1]. Prefrontal involvement in schizophrenia and depression has been identified by various brain imaging methods measuring cerebral blood flow at rest and during task load.

In schizophrenia, Ingvar and Franzén [2] reported that in a study using the 133Xenon clearance method, the resting cerebral blood flow in chronic schizophrenics is low in the prefrontal region relative to the hemisphere mean. Franzén and Ingvar [5] also revealed that chronic schizophrenics did not show a flow increase during mental tasks, indicating that they manifest both resting state and activation deficits in the prefrontal region. Hypofrontality in schizophrenia during activated states was confirmed by other studies using single-photon-emission computed tomography (SPECT) [6,7], functional magnetic resonance imaging (fMRI) [8,9,10], and near-infrared spectroscopy (NIRS) [11,12,13,14,15,16]. The employed tasks include verbal fluency, decision making, the viewing of an emotional film, the Wisconsin Card Sorting Test, the Stroop task, and the random-number-generation task. These studies indicate that the data from schizophrenic subjects exhibit a decreased response of cerebral blood flow to task load.

In comparison with the data during the prefrontal activation, during the resting condition, results are inconsistent, and the results vary depending on the study. Some studies have replicated hypofrontality [3,17]. Additionally, in other studies, the opposite results were presented. Soyka et al. [18] showed that a hypermetabolic pattern of the prefrontal activity in the resting state was observed in a positron emission tomography (PET) study on schizophrenia. In that study, a hyperfrontal metabolic pattern was found using the occipital cortex as the reference region. Catafau et al. [6] also reported in a SPECT study that, under the resting condition, schizophrenic patients had significantly higher blood flow in the prefrontal brain.

Blood flow alterations in the prefrontal region have also been reported in depression, both during the activated and resting conditions. Decreased activation during task load was reported in the studies using 133Xenon SPECT [19] and NIRS [20,21,22,23,24]. On the contrary, a study by Walter [25] reported an increased activation of the frontal cortex in depression.

Regarding the resting-state measurements in depression, Buchsbaum et al. [3], in their PET study, reported that patients both with schizophrenia and with affective illness showed a diminished anteroposterior gradient, indicating relative hypofrontal activity. Resting hypofrontality in depression was also replicated in the studies using SPECT [26,27]. It is also suggested that the pattern of hemodynamic dysfunction in depression is different from that in schizophrenia [19,20,28].

These previous data indicate that the prefrontal blood flow activation during task load is largely reduced in both schizophrenia and depression. Reactivity of the prefrontal brain to applied tasks may be diminished in both disorders. Reduction in resting-state blood flow is also reported in both disorders, but hyperfrontality has been reported in schizophrenia. Inconsistent results may be due to methodological differences. The resting-state data in the prefrontal region were usually analyzed with reference to those in the other areas of the brain, such as the posterior part of the brain or the mean hemisphere data. Conventional methods have reported blood flow alterations in the prefrontal region in schizophrenia and depression relative to the reference area, but the distinction between these two disorders is difficult [3]. The present study aimed to investigate the absolute resting prefrontal blood volume with near-infrared time-resolved spectroscopy (NIR-TRS), which measures the absolute hemoglobin concentrations [29,30], and examine if schizophrenia and depression can be differentiated.

NIRS has been widely employed to examine the brain dysfunction of schizophrenia and depression, as introduced above. Most of these previous studies used the continuous-wave (CW) type of NIRS (CW-NIRS). CW-NIRS is useful in detecting the changes in blood flow, but its utility is limited by the lack of measurement for the absolute value of hemoglobin concentration due to unknown pathlengths [29,30]. NIR-TRS, on the other hand, estimates absorption coefficients to calculate the absolute hemoglobin concentration, and was utilized in the present study to assess the prefrontal blood flow in schizophrenia and major depressive disorder (MDD).

In our previous study [30] using NIR-TRS, total hemoglobin concentration in the anterior prefrontal cortex of schizophrenia was found to be reduced. The present study extended this research to examine the oxygenated and deoxygenated hemoglobin concentration, not only in schizophrenia but also in MDD. The data were also normalized by systemic hemoglobin concentration to assess the changes in blood volume. The present study investigated the usefulness of NIR-TRS measurement in the anterior prefrontal brain for detecting hypofrontality in schizophrenia and making a differential diagnosis of schizophrenia and depression.

## 2. Materials and Methods

### 2.1. Subjects

The subjects were 12 schizophrenic patients (age, 39.6 ± 13.4 years, mean ± s.d.; 6 males and 6 females), and age- and gender-matched patients with MDD (*n* = 15, 39.1 ± 8.9 years; 7 males and 8 females) and control subjects (*n* = 15, 37.9 ± 12.0 year; 6 males and 9 females). The age of onset and the duration of illness were 19.6 ± 6.1 years and 20.0 ± 11.4 years, respectively, in schizophrenic patients, and 30.6 ± 9.7 years and 8.5 ± 4.6 years, respectively, in depression. The schizophrenic and MDD patients were diagnosed based on the criteria of DMS-5 [4] and treated conventionally at Maynds Tower Mental Clinic with anti-psychotics (38.8 ± 80.0 mg chlorpromazine-equivalent) and anti-depressants (25.2 ± 34.6 mg, fluvoxamine-equivalent), respectively [31]. Written informed consent to participate in the present study was obtained from all subjects.

### 2.2. Measurement Protocol

During the NIR-TRS measurement, the subject was seated on a chair in the eyes-open condition. The eyes were open to maintain the awake state. The subject was instructed to look at the wall in front of the patient and to not tilt the head down during the measurement because the head position can affect the NIR-TRS data. 

A single-channel time-resolved spectroscopy instrument (TRS-10, Hamamatsu Photonics, Hamamatsu, Japan) [32] was used to measure hemoglobin concentrations in the anterior prefrontal region during the resting state. The incident and the detecting light guides were placed at the forehead 3 cm horizontally apart, with the center being set at Fpz of the international 10–20 system. The interval between the incident and the detecting guides was 3 cm following our previous study with successful measurements of hemoglobin concentration using the same system [30]. The light guides were securely attached to the skin surface using adhesive tape and a headband. The data were recorded for 1 min, and the averaged hemoglobin concentration was used for analysis. A detailed description of the measurement methods is provided below.

In the present study, the brain area under Fpz was assessed, which covers the bilateral anterior frontal areas, including Broadman A10 and the surrounding regions. The anterior frontal areas were the target of our previous studies using CW-NIRS and NIR-TRS on schizophrenia and depression [13,30,33], and are considered to be involved in the pathophysiology of these disorders. The anterior frontal areas were also assessed in various NIRS studies on schizophrenia and depression [11,12,13,14,15,16,20,21,22,23,24], indicating the importance of recording at these areas. The present findings would not represent the whole prefrontal areas, but may indicate the usefulness of the anterior prefrontal data for making the diagnosis and evaluating the symptoms of schizophrenia and depression. The protocol of the present study was approved by the institutional review board at Shizuoka Saiseikai General Hospital.

### 2.3. Near-Infrared Time-Resolved Spectroscopy

The TRS-10 system consists of three pulsed laser diodes with different wavelengths (759, 796, and 835 nm) having a duration of around 100 ps at the repetition rate of 5 MHz, a high-speed photomultiplier tube for single-photon detection, and a circuit for time-resolved measurement based on a time-correlated single-photon counting (TCSPC) method. It takes 1 s to obtain the time-of-flight (TOF) distribution of photons for each wavelength. The TOF distributions with adequate signal-to-noise ratios are acquired by summation of signals for 10 s. Single data acquisition takes 1 s, and is repeated 10 times to calculate one hemoglobin concentration score. Six scores are averaged in 1-min measurements. The present method is based on our previous study using the same system [30], and is improved by increasing the repetition number from 5 to 10. The instrument response function (IRF) is measured by placing the incident light guide (200 μm diameter, NA = 0.25) opposite the detecting fiber (bundle fiber, 3 mm diameter, NA = 0.21) with a neutral density filter between them. The IRF FWHM (full width half maximum) is about 150 ps at each wavelength.

The absorption (μ_a_) and reduced scattering coefficients (μ_s_) are estimated by fitting the TOF distribution, which is derived from the analytical solution of the photon diffusion equation with the extrapolated boundary condition [34] and convoluted by the IRF, into the observed TOF distribution (time range of 0–5400 ps). In the fitting procedure, a weighted least-squares fitting method based on the Levenberg–Marquardt Method is used. The refractive index in the human head is assumed to be 1.37.

The photons emitted from the incident light guide enter the brain, are scattered by the brain structures, are partially absorbed by hemoglobin molecules, and then leave the brain to reach the detecting light guide. The optical pathlength is the distance that the photons have moved in the brain and is expressed as the following equation:Optical Pathlength = (Light speed/Refractive index) × Mean TOF(1)
where:
Light Speed = 0.29979 mm/psRefractive Index in the brain = 1.37Mean TOF = ∫ t f(t)dt/∫ f(t)dtDistribution of TOF = f(t)

### 2.4. Calculation of Absolute Value of Cerebral Hemoglobin Concentrations

The concentration of oxygenated (oxy-Hb) and deoxygenated hemoglobin (deoxy-Hb) are expressed by the following equation on the assumption that background absorption is only due to water:μ_aλn_ = ln10(ε_λn_ [oxy-Hb] + ε’_λn_ [deoxy-Hb]) + μ_a water λn_ [water](2)
where ελ_n_ and ε’λ_n_ are the molar absorption coefficients of oxy-Hb and deoxy-Hb at λ_n_ (λ_1_ = 759 nm, λ_2_ = 796 nm, λ_3_ = 835 nm), respectively, and μ_a_ water λ_n_ is the absorption coefficient of water at each wavelength. Oxy-Hb and deoxy-Hb are the molar concentrations, and [water] is the volume fraction of water content, which is considered a constant (60%) [35]. The least-square technique is employed to calculate oxy-Hb and deoxy-Hb. After the water absorption at each wavelength is subtracted from the μ_a_λ_n_, a normal equation is derived from Equation (2). Then, oxy-Hb and deoxy-Hb are calculated from this normal equation. The summation of oxy-Hb and deoxy-Hb provides total-Hb. Oxygen saturation of hemoglobin (SO_2_) is calculated as [oxy-Hb]/[total-Hb].

### 2.5. Ratios to Systemic Blood Hemoglobin Concentration

The systemic blood hemoglobin concentration (blood-Hb) was measured conventionally in the venous blood samples with the Sodium Lauryl Sulfate method [36]. The prefrontal hemoglobin concentrations were divided by blood-Hb data in order to normalize them with respect to the systemic hemoglobin concentration. The hemoglobin concentration in the brain should be affected by that in the body. Calculation of the ratios to systemic hemoglobin concentration can enhance the detection of changes originated in the brain. The unit of the normalized values was arbitrary because that of the prefrontal hemoglobin concentration was microM, and that of the venous hemoglobin concentration was g/dL. This normalization procedure for oxy-Hb, deoxy-Hb, and total-Hb enables the assessment of the contribution of blood volume on analyzing the data.

### 2.6. Statistical Analysis

Statistical analysis was conducted using commercial software (Prism ver. 8, GraphPad, San Diego, CA, USA). The hemoglobin concentration data, optical pathlengths, SO_2_, and age were analyzed by one-way ANOVA with post hoc Tukey’s multiple comparison test. The effect size and the 95% confidence interval (CI) of difference were also used to confirm the findings. The gender ratio was checked by the chi-squared test. The correlation between the NIR-TRS data and the medication dosage was examined using Spearman correlation coefficients.

## 3. Results

The mean and standard deviation of the NIR-TRS parameters, in addition to the F values and *p*-values in a one-way ANOVA for the control, schizophrenic, and depression data, are presented in Table 1. Oxy-Hb, total-Hb, oxy-Hb/blood-Hb, deoxy-Hb/blood-Hb, and total-Hb/blood-Hb showed statistically significant group effects (Table 1). Blood-Hb, SO_2_, and the optical pathlengths for the near-infrared light with three wavelengths did not show group differences.

Box plot data in three groups, together with *p*-values in post hoc Tukey’s multiple comparison test, are shown in Figure 1.

In schizophrenia, total-Hb in schizophrenic patients was lower than that in the control (*p* = 0.047, mean difference = 12.34, effect size = 0.883, CI of difference = 0.1601 to 24.51). When the ratio of prefrontal hemoglobin concentration to blood-Hb was used, oxy-Hb/blood-Hb (*p* = 0.018, mean difference = 0.6590, effect size = 0.951, CI of difference = 0.09897 to 1.219), deoxy-Hb/blood-Hb (*p* = 0.028, mean difference = 0.3630, effect size = 0.959, CI of difference = 0.03416 to 0.6919), and total-Hb/blood-Hb (*p* = 0.012, mean difference = 1.021, effect size = 1.032, CI of difference = 0.2012 to 1.841) all showed a significant reduction compared to the control.

In MDD, oxy-Hb, deoxy-Hb, and total-Hb, together with their ratios to blood-Hb, were not statistically different from the control data (*p* > 0.05). 

When comparing the data between schizophrenia and depression, oxy-Hb (*p* = 0.011, mean difference = −10.88, effect size = 1.242, CI of difference = −19.56 to −2.192), total-Hb (*p* = 0.019, mean difference = −14.23, effect size = 1.161, CI of difference = −26.41 to −2.057), oxy-Hb/blood-Hb (*p* = 0.003, mean difference = −0.8110, effect size = 1.560, CI of difference = −1.371 to −0.2510), and total-Hb/blood-Hb (*p* = 0.008, mean difference = −1.070, effect size = 1.396, CI of difference = −1.890 to −0.2505) in schizophrenia were significantly lower than the depression data.

In schizophrenia and MDD, Spearman correlation coefficients showed that oxy-Hb, deoxy-Hb, total-Hb, oxy-Hb/blood-Hb, deoxy-Hb/blood-Hb, total-Hb/blood-Hb, and SO_2_ were not correlated with the chlorpromazine-equivalent dose of anti-psychotic medication and the fluvoxamine-equivalent dose of anti-depressant medication (*p* > 0.05).

## 4. Discussion

### 4.1. Prefrontal Hemoglobin Concentration in the Healthy Control Assessed by NIR-TRS

In the present study, the absolute value of the hemoglobin concentration was measured in the anterior part of the prefrontal cortex in the healthy control subjects using NIR-TRS. NIR-TRS estimates optical absorption coefficients and pathlengths for the emitted photons. Based on these parameters, the absolute values of hemoglobin concentration are obtained for the area where the photons have moved. In the present study, using a single-channel type of NIR-TRS instrument, the midpoint between the incident and detecting probes with the inter-probe interval of 3 cm was set at Fpz. The present result in the healthy control subjects was 63.5 ± 14.1 microM with optical pathlengths of about 20 cm (Table 1). It is speculated that, in the present system, the emitted photons enter the brain, are scattered inside it within the depth of several centimeters of the cortex, and leave to reach the detector after moving the distance of about 20 cm [29].

The observed data were comparable with but lower than our previous measurements in control subjects at the left prefrontal (averaged total-Hb; 68.9 microM) and the right prefrontal (70.4 microM) areas [30]. The difference between the present and previous data may be due to the anatomical differences in the recording sites. In the present recording, bilateral frontopolar areas and sagittal sulcus were directly under the probes. In our previous study, more lateral parts of the prefrontal cortex may be involved.

Sakatani and colleagues showed, using NIR-TRS, that the averaged total-Hb concentrations at the left and right prefrontal areas were 57.3 and 57.4 microM, respectively, in healthy female subjects with a mean age of 46.8 years [37], and were 55.17 and 58.15 microM, respectively, in normal male and normal female subjects with a mean age of 73.4 years [38]. Their hemoglobin concentration data were lower than ours, suggesting that age can affect the hemoglobin concentration of the brain. SO_2_ was 66.7 and 66.6% in the former group and 67.4 and 67.7% in the latter group, and was comparable to our data. Their NIR-TRS instrument also uses three photodiodes with wavelengths that are almost the same as ours [38]. Future studies are warranted to examine the NIR-TRS data in healthy subjects with respect to age.

### 4.2. Prefrontal Hemoglobin Concentration in Schizophrenia

The present study showed that the hemoglobin concentration (total-Hb) in the anterior prefrontal brain of schizophrenic patients is significantly reduced compared to the control. The result confirmed our previous finding on schizophrenia reporting reduced total-Hb [30]. The present findings are consistent with the previous studies using other methods, including SPECT, PET, and fMRI, showing a reduced prefrontal blood flow in schizophrenia [2,3,5,17]. SPECT and PET use radioactive substances inhaled or injected into the blood vessels. NIR-TRS uses only infrared light and measures intrinsic hemoglobin concentration in the brain to analyze cerebral blood flow. In analyzing SPECT data, arterial measurement of radioactive substances is necessary to calculate the absolute values of brain concentration [39]. The NIR-TRS method is simple and can be conducted in a regular room that is not specialized for the measurement. Strict head restriction is not required in NIR-TRS, and the measurement at the resting state only takes several minutes. It could be possible to use NIR-TRS in daily clinical practices to diagnose schizophrenia.

The present study also revealed that the saturation of hemoglobin in the brain of schizophrenia is not different from that of control subjects in spite of a decrease of total-Hb in schizophrenia, suggesting that the prefrontal oxygen consumption is reduced in schizophrenia. The present study further analyzed the normalized data and showed that both oxy-Hb/blood-Hb and deoxy-Hb/blood-Hb were significantly reduced in schizophrenia in addition to total-Hb/blood-Hb. The results support the view that the cerebral blood volume itself is lowered in the prefrontal brain area in schizophrenia patients. Optical pathlengths in NIR-TRS for schizophrenic patients were not different from those in the control (Table 1). Differences in the light-scattering condition inside the prefrontal brain should not be the cause of the blood flow reduction in schizophrenia. Structural changes in the schizophrenic brain have been suggested in previous studies and can underlie the reduced blood volume [40]. Future studies are necessary to examine the relation between structural changes in the prefrontal cortex of schizophrenia and blood volume reduction.

Previous studies on resting blood flow in the prefrontal brain using SPECT, PET, and fMRI are inconsistent and have shown both hypofrontality and hyperfrontality. These methods mostly evaluate the amount of blood flow by analyzing its distribution in the brain. Schizophrenia may accompany the absolute reduction in the blood volume in the prefrontal area in addition to the disturbed blood flow distribution in the brain.

### 4.3. Prefrontal Hemoglobin Concentration in Depression

In depression, NIR-TRS did not reveal significant differences from the control, including total-Hb, oxy-Hb, deoxy-Hb, total-Hb/blood-Hb, oxy-Hb/blood-Hb, deoxy-Hb/blood-Hb, and SO_2_. The absolute blood volume in the anterior prefrontal brain may not be affected in depression. 

Previous reports using SPECT, PET, and fMRI indicated a reduction in the prefrontal blood flow. The discrepancy from the present results may come from the methodological differences. SPECT, PET, and fMRI studies mostly show the changes in blood flow distribution. Hypofrontality with these methods means that the blood flow in the anterior part of the brain is lower than that in the posterior part. Depression may be accompanied by a normal prefrontal blood volume but by an altered blood flow distribution. Future studies are warranted to verify this issue using both NIR-TRS and SPECT, PET, or fMRI.

### 4.4. Assessment of Prefrontal Blood Volume in Schizophrenia and Depression Using NIR-TRS

As described in the Introduction section, the prefrontal brain is engaged in various functions, including the planning, decision, and execution of behavior, and is related to both schizophrenia and depression [1]. The present results indicate that the absolute blood volume in the prefrontal brain is reduced in schizophrenia and is not reduced in MDD. The differences in hemoglobin concentrations, in addition to their data normalized by systemic blood hemoglobin concentrations between these disorders, are statistically significant (Figure 1). In addition to other imaging methods, NIR-TRS measurement of the anterior prefrontal blood volume will be informative in not only evaluating the dysfunction of the prefrontal cortex, but also making the differential diagnosis of schizophrenia and MDD. NIR-TRS can be a tool to assess the blood flow dysfunction in the brain of schizophrenia and depression, in addition to the conventional methods including SPECT, PET, fMRI, and the continuous-wave type of NIRS.

In the present study with NIR-TRS, optical pathlengths were obtained and showed inter-subject variation, with the standard deviation being around 10% of the mean, as indicated in Table 1. In CW-NIRS, photons are emitted continuously. The pathlength cannot be known and is given a fixed value in the calculation of hemoglobin concentration. The difference between the real pathlength and the fixed value in each subject leads to an erroneous calculation of absolute hemoglobin concentration in the brain. In NIR-TRS, on the other hand, the accurate optical pathlength is obtained on every measurement, supporting the calculation of absolute hemoglobin concentration based on the absorption coefficients. Therefore, NIR-TRS has an advantage when absolute hemoglobin concentration and blood volume are assessed, as in the case of the present study.

### 4.5. Limitation

In the present study, a single-channel NIR-TRS device was used to mainly measure the anterior part of the prefrontal cortex. Lateral, medial, and basal parts of the prefrontal cortex may be differently involved in the pathophysiology of schizophrenia and depression [1], and should be assessed in future research. The present study also employed a small number of patients and controls, and the results are considered preliminary. The analyzed data may contain some errors related to non-significant results due to the small sample size, although the effect sizes and the confidence intervals of the data support the validity of the present findings. Future studies on a larger number of participants are required to consolidate the present findings. The schizophrenic and MDD patients in the present study were chronic, with the mean duration of illness being 20.0 and 8.5 years, respectively. The effects of age, duration of illness, and the medication dosage, in addition to the relation to clinical symptoms, should be further analyzed in the studies with larger sample sizes. The assessment of the prefrontal blood volume in other psychiatric disorders, including bipolar disorder and neurodevelopmental disorder, will be interesting.

## 5. Conclusions

NIR-TRS showed an anterior prefrontal blood volume reduction in schizophrenia but not in MDD. It is suggested that NIR-TRS measurement in the anterior prefrontal brain is useful for detecting the hypofrontality in schizophrenia and making a differential diagnosis of schizophrenia and depression.

## Figures and Tables

**Figure 1 sensors-22-01594-f001:**
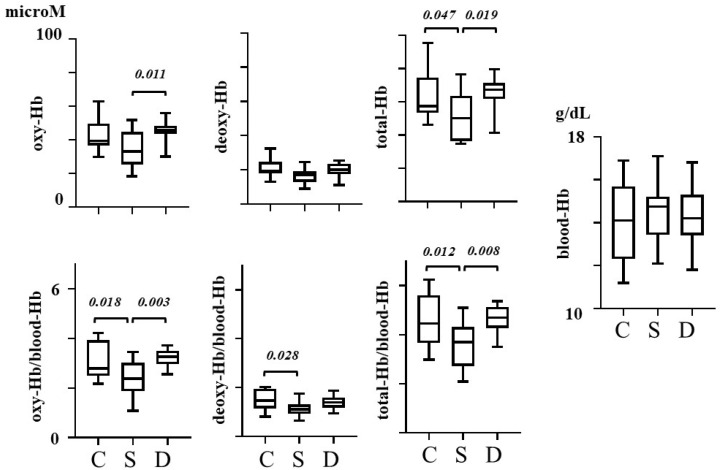
Box plot presentation of the data in control (C), schizophrenia (S), and depression (D) groups with *p*-values of Tukey’s multiple comparison test. Horizontal lines connecting two groups indicate the statistically significant difference between the data averages. *p*-values of Tukey’s multiple comparison test are shown above the line.

**Table 1 sensors-22-01594-t001:** Mean (s.d.) data in the control, schizophrenia, and depression groups with F and *p*-values of one-way ANOVA. *p*-values are presented in italic when ANOVA showed significant effects. oxy-Hb, oxygenated hemoglobin concentration; deoxy-Hb, deoxygenated hemoglobin concentration; total-Hb, total hemoglobin concentration; blood-Hb, systemic venous blood hemoglobin concentration; SO_2_, oxygen saturation of hemoglobin.

		Control		Schizophrenia		Depression		One-Way ANOVA	
		Mean	(s.d.)	Mean	(s.d.)	Mean	(s.d.)	F(2,39)	*p*
oxy-Hb	microM	42.8	(10.0)	34.7	(10.4)	45.6	(7.1)	1.84	*0.013*
deoxy-Hb	microM	20.7	(5.2)	16.4	(4.6)	19.8	(4.1)	1.06	0.058
total-Hb	microM	63.5	(14.1)	51.1	(13.9)	65.4	(10.7)	2.76	*0.016*
oxy-Hb/total-Hb (SO_2_)	%	67.4	(4.8)	67.6	(6.5)	69.9	(2.6)	0.68	0.289
blood-Hb	g/dL	14.1	(1.7)	14.6	(1.6)	14.1	(1.3)	0.32	0.691
oxy-Hb/blood-Hb		3.06	(0.70)	2.40	(0.69)	3.21	(0.35)	0.09	*0.003*
deoxy-Hb/blood-Hb		1.50	(0.42)	1.14	(0.34)	1.40	(0.27)	0.07	*0.032*
total-Hb/blood-Hb		4.56	(1.03)	3.54	(0.95)	4.61	(0.58)	0.15	*0.005*
pathlength (759 nm)	cm	20.4	(2.3)	21.4	(2.9)	20.8	(1.9)	0.48	0.585
pathlength (796 nm)	cm	20.3	(2.2)	21.4	(3.0)	20.8	(1.6)	0.41	0.483
pathlength (835 nm)	cm	19.2	(2.0)	20.0	(2.8)	19.5	(1.5)	0.39	0.598

## Data Availability

The data that support the findings of this study are available on request from the corresponding author. The data are not publicly available due to privacy and ethical restrictions.

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
