# Peer review of "Near-Infrared Time-Resolved Spectroscopy Shows Anterior Prefrontal Blood Volume Reduction in Schizophrenia but Not in Major Depressive Disorder"

_sensors, 2022, doi:10.3390/s22041594_

Round 1

Reviewer 1 Report

Review the paper entitled “Near-infrared Time-Resolved Spectroscopy Shows Prefrontal Blood Volume Reduction in Schizophrenia but not in Major Depressive Disorder.

The author utilized a single-channel near-infrared time-resolved spectroscopy (NIR-TRS) to study the hemodynamic activity of patients with Schizophrenia but not in Major Depressive Disorder. The authors reported that the total-Hb and total-Hb/blood-Hb, oxy-Hb/blood-Hb, and deoxy-Hb/blood-Hb in Schizophrenia were reduced.   However, total-Hb, total-Hb/blood-Hb, oxy-Hb, and oxy-Hb/blood-Hb of patients with depression were higher than schizo-27 phrenia and were not different from the control. 

The present result may be clinically significant if a single channel fNIRS can differentiate patients with Schizophrenia from normal controls. It may potentially develop into a screening test for individuals at risk for psychotic disorders.

My major question is the non-significant different result of individuals with depression. To take it as a reliable finding, one may question whether there may be errors (small sample size, heterogenous within-group) to affect the result. It is recommended that the author conduct a power analysis and describe in more detail the characteristic of the patients with depression (e.g., age of onset, duration of illness)

What is the difference between fNIRS and Near-infrared Time-Resolved Spectroscopy?   Although the author stated, “Most of these previous studies used continuous 93 waves (CW) type of NIRS (CW-NIRS). CW-NIRS is useful in detecting the changes in blood 94 flow. Still, its utility is limited by the lack of measurement for the absolute value of hemo-95 globin concentration due to unknown path lengths. NIR-TRS, on the other hand, 96 measures pathlengths to calculate the absolute hemoglobin concentration”. I am not sure about the advantage of Time-resolved with the more commonly used fNIRS.    Specifically, why is it significant to calculate the absolute concentration?

Author Response

Dear Reviewer 1

Thank you for the valuable comments and suggestions. I am glad to read your comment ‘The present result may be clinically significant if a single channel fNIRS can differentiate patients with Schizophrenia from normal controls. It may potentially develop into a screening test for individuals at risk for psychotic disorders.’

Following are the responses to your comments and suggestions. We modified the manuscript following them. I believe that the manuscript is improved, and hope that it is acceptable for publication.

Comment 1: My major question is the non-significant different result of individuals with depression. To take it as a reliable finding, one may question whether there may be errors (small sample size, heterogenous within-group) to affect the result. It is recommended that the author conduct a power analysis and describe in more detail the characteristic of the patients with depression (e.g., age of onset, duration of illness)

Response: Thank you for the important comment. We presented 95% confidence intervals for the analysis of the data to confirm the results (p5, line214-215, line228-240). We also added the age of onset and duration of illness for the patients to clarify the clinical profiles (p3, line116-118; p9, line368-369).

Comment 2: What is the difference between fNIRS and Near-infrared Time-Resolved Spectroscopy?   Although the author stated, “Most of these previous studies used continuous waves (CW) type of NIRS (CW-NIRS). CW-NIRS is useful in detecting the changes in blood flow. Still, its utility is limited by the lack of measurement for the absolute value of hemoglobin concentration due to unknown path lengths. NIR-TRS, on the other hand, measures pathlengths to calculate the absolute hemoglobin concentration”. I am not sure about the advantage of Time-resolved with the more commonly used fNIRS. Specifically, why is it significant to calculate the absolute concentration?

Response: Thank you for the valuable comment. Hemoglobin concentration is obtained using the optical pathlengths. Optical pathlengths show inter-subject variation with the standard deviation being around 10 % of the mean as presented in Table 1. In CW-NIRS, photons are emitted continuously. So, the pathlength cannot be known and is given a fixed value, such as 30 cm, in the calculation of hemoglobin concentration. The difference between the real pathlength and the fixed value in each subject leads to erroneous calculation of absolute hemoglobin concentration in the brain. In NIR-TRS, on the other hand, the real optical pathlength is obtained on every measurement, enabling the accurate calculation of absolute hemoglobin concentration. Therefore, NIR-TRS has advantage when absolute hemoglobin concentration and blood volume are assessed, as in the case of present study. The above statements are added to the manuscript (p8, line350-359).

Reviewer 2 Report

General comments

In general, the article evaluates an important topic: prefrontal blood volume differs among schizophrenia, major depressive and healthy control subjects. This article measured the concentrations of oxy-Hb, deoxy-Hb, total-Hb in the forehead among schizophrenic, major depressive, and control subjects by using the near-infrared time-resolved spectroscopy (NIR-TRS). The results are informative, however, I do have some concerns regarding how the results can be translated to the clinical settings.

Major concerns:

  1. Please verify that 1 min measurement of NIRS-TRS is enough for further analysis of absolute concentration of oxy-Hb, deoxy-Hb, and total-Hb.
  2. Please explain why Fpz was selected in the present study? Since few NIRS-TRS studies measured this brain area, how the results from current study related to the literature?
  3. Are there any differences in the concentration of oxy-Hb between different brain regions in the prefrontal cortex among different subject groups? Please justify how Fpz is a representative of the blood flow/volume of the prefrontal brain?
  4. Please verify the optode space used in the present study? Why 3 cm was used in the present study?
  5. Please give detailed information about how the blood-Hb was measured?
  6. Please validate the ratio to systemic blood-Hb concentration.

Minor concerns

Page 1, line 40-43. It would be good to cite some references to support these two statements.

Page 1, line 44. Please check the grammar of this sentence: “the latter…” is a not a complete sentence. 

Page 2,line 60, please specify, was it “the data from schizophrenic subjects”?

Page 2, line 62, I would be recommend to replace “And” with “However”.

Page 2, line 72. “an increased activation in depressed subjects?”

Page 2, line 74. Please check the grammar of this sentence.

Page 3, line 104-107. I would recommend to remove these sentences regarding the results, and add more information about the purpose/hypothesis of the present study.

There are many grammatical errors in this manuscript. It makes the article a little distracting to read.  The authors should ask someone else who is a fluent native English writer to proofread.

Author Response

Dear Reviewer 2

Thank you for the valuable comments and suggestions. I am glad to read your comment ‘In general, the article evaluates an important topic: prefrontal blood volume differs among schizophrenia, major depressive and healthy control subjects.’ And I also agree that ‘the results should be translated to the clinical settings.’ Following are the comments and our responses.

Major concerns:

Comment 1. Please verify that 1 min measurement of NIRS-TRS is enough for further analysis of absolute concentration of oxy-Hb, deoxy-Hb, and total-Hb.

Response: Thank you for the important question. Single data acquisition takes 1 s, and is repeated 10 times to calculate one hemoglobin concentration score. Six scores are averaged in 1 min measurement. The present method is based on our previous study using the same system [30], and is improved by increasing the repetition number from 5 to 10. The above description is added to the method section (p4, line158-161).

Comment 2. Please explain why Fpz was selected in the present study? Since few NIRS-TRS studies measured this brain area, how the results from current study related to the literature?

Response: Thank you for the question. In our previous studies on schizophrenia and depression (Shinba et al., 2004, Hoshi et al., 2006, Shinba et al., 2017), the anterior parts of the frontal lobe (Fp1-F7, Fp2-F8) was assessed. Recent CW-NIRS studies also focused on the anterior frontal dysfunction. These descriptions are added to the manuscript (p3, line140-145).

Comment 3. Are there any differences in the concentration of oxy-Hb between different brain regions in the prefrontal cortex among different subject groups? Please justify how Fpz is a representative of the blood flow/volume of the prefrontal brain?

Response: Thank you for the important comment. The present finding with recording at Fpz area is not the representative of the blood volume of the whole prefrontal brain areas. Recordings at other prefrontal areas may show different results. The present finding is for the anterior prefrontal blood volume. The title of the manuscript is changed as well as the manuscript (Title; p3, line145-147).

Comment 4. Please verify the optode space used in the present study? Why 3 cm was used in the present study?

Response: Thank you for the question. For the present TRS device, 3 cm optode space is adequate to measure the hemoglobin concentration in the brain, and has been used in our previous study (Hoshi et al., 2006). These descriptions are added to the manuscript (p3, line133-135).

Comment 5. Please give detailed information about how the blood-Hb was measured?

Response: Thank you for the comment. The reference is cited in the revised manuscript (p4, line202).

Comment 6. Please validate the ratio to systemic blood-Hb concentration.

Response: The hemoglobin concentration in the brain should be affected by that in the body. Calculation of the ratios to systemic hemoglobin concentration can enhance the detection of changes originated in the brain. This description is added to the manuscript (p4, line204-207)

Minor concerns

Comment 1. Page 1, line 40-43. It would be good to cite some references to support these two statements.

Response: The appropriate citations are added to the manuscript (p1, line42, 44).

  1. Page 1, line 44. Please check the grammar of this sentence: “the latter…” is a not a complete sentence.

Response: Thank you for the comment. We asked the English editing service of MDPI to improve English of this manuscript. The certificate is also uploaded.

  1. Page 2,line 60, please specify, was it “the data from schizophrenic subjects”?

Response: Thank you for your comment. the description `the data from schizophrenic subjects’ is used in the manuscript (p2, line59).

  1. Page 2, line 62, I would be recommend to replace “And” with “However”.

Response:  Thank you for the comment. The English editing service put ‘Additionally’ here (p2, line63).

  1. Page 2, line 72. “an increased activation in depressed subjects?”

Response: Thank you for correcting the grammatical error (p2, line73)

  1. Page 2, line 74. Please check the grammar of this sentence.

Response: Thank you for the comment. We asked the English editing service of MDPI to improve English of this manuscript (p2, line75).

  1. Page 3, line 104-107. I would recommend to remove these sentences regarding the results, and add more information about the purpose/hypothesis of the present study.

Response: Thank you for the comment. We removed the sentences regarding the results and clarified the purpose of the present study (p3, line105-108).

  1. There are many grammatical errors in this manuscript. It makes the article a little distracting to read. The authors should ask someone else who is a fluent native English writer to proofread.

Response: Thank you for the comment. We asked the English editing service of MDPI to improve English of this manuscript. The certificate is also uploaded.

Reviewer 3 Report

General remarks:
a.    It is necessary, in my opinion, to underline the fact that described results were obtained for a small group of patients, so they should be treated as preliminary.
b.    Figure 1 presents obtained data in a naïve way. Have authors tried to use so called ‘box plot’ presentation?
c.    Authors use terms, which are not commonly accepted or have other meaning than intended (e.g. measures the optical absorption coefficient and calculates the absolute concentrations, measures pathlengths to calculate, was checked by Sperman correlation coefficients, photons have traveled, in a larger population, just to mention a few examples).
d.    Some minor mistakes can be found in the reference section.

Below I list some lines with statements containing errors or not clear to me:
Lines: 26-29, 96-97, 148-150, 162, 170, 187-189, 245-246, 252-253, 262-263, 279-280, 293-294, 317-318.

Author Response

Dear Reviewer 3

Thank you for the valuable comments and suggestions. The manuscript is modified according to them. Following are the responses to your comments and suggestions. I believe that the manuscript is improved, and hope that it is acceptable for publication.

General remarks:

Comment a.    It is necessary, in my opinion, to underline the fact that described results were obtained for a small group of patients, so they should be treated as preliminary.

Response: Thank you for the important comment. The word ‘preliminary’ is added to the discussion section of the manuscript (p9, line366).

Comment b.    Figure 1 presents obtained data in a naïve way. Have authors tried to use so called ‘box plot’ presentation?

Response: Thank you for the suggestion. Box plots are used in the revised manuscript (Figure 1).

Comment c.    Authors use terms, which are not commonly accepted or have other meaning than intended (e.g. measures the optical absorption coefficient and calculates the absolute concentrations, measures pathlengths to calculate, was checked by Sperman correlation coefficients, photons have traveled, in a larger population, just to mention a few examples).

Response: The descriptions are modified and clarified in the revised manuscript (lines21, 174, 218, 265, 271, 371).

Comment d.    Some minor mistakes can be found in the reference section.

Response: The reference list is checked again and the errors were removed (lines422, 427).

Below I list some lines with statements containing errors or not clear to me:

Lines: 26-29, 96-97, 148-150, 162, 170, 187-189, 245-246, 252-253, 262-263, 279-280, 293-294, 317-318.

Response: Thank you for your comment. We asked the English editing service of MDPI to improve English of this manuscript and to clarify the sentences.

Round 2

Reviewer 1 Report

You did not do the power analysis and address the question of error related to non-significant difference of the results. 

Author Response

Dear Reviewer 1

Thank you for the important comment, ‘You did not do the power analysis and address the question of error related to non-significant difference of the results.’

Response: For the power analysis, we calculated effect sizes (p5 line212). The effect sizes were large enough to support the findings, and presented in the result section (p5 lines227, 229, 230, 232, 237, 238, 239, and 240). We also added the following sentence in 4.5 Limitation subsection of Discussion (p9 line365) ‘The analyzed data may contain some errors related to non-significant results due to the small sample size, although the effect sizes and the confidence intervals of the data support the validity of the present findings.’

We hope that this will be the answer to your comment.

Sincerely yours,

Toshikazu Shinba